# The Urgent Need for Cardiopulmonary Fitness Evaluation among Wildland Firefighters in Thailand

**DOI:** 10.3390/ijerph20043527

**Published:** 2023-02-16

**Authors:** Jinjuta Panumasvivat, Wachiranun Sirikul, Vithawat Surawattanasakul, Kampanat Wangsan, Pheerasak Assavanopakun

**Affiliations:** 1Department of Community Medicine, Faculty of Medicine, Chiang Mai University, Chiang Mai 50200, Thailand; 2Center of Data Analytics and Knowledge Synthesis for Health Care, Chiang Mai University, Chiang Mai 50200, Thailand

**Keywords:** wildland firefighter, fitness for duty, cardiopulmonary fitness, aerobic capacity, occupational health and safety

## Abstract

Wildland firefighting is a high-risk occupation. The level of cardiopulmonary fitness can indicate whether wildland firefighters are ready to perform their job duties. This study’s objective was to determine wildland firefighters’ cardiopulmonary fitness using practical methods. This cross-sectional descriptive study aimed to enroll all 610 active wildland firefighters in Chiang Mai. The participants’ cardiopulmonary fitness was assessed using an EKG, a chest X-ray, a spirometry test, a global physical activity questionnaire, and the Thai score-based cardiovascular risk assessment. The NFPA 1582 was used to determine “fitness” and “job restriction”. Fisher’s exact and Wilcoxon rank-sum tests were used to compare cardiopulmonary parameters. With a response rate of 10.16%, only eight wildland firefighters met the cardiopulmonary fitness requirements. Eighty-seven percent of participants were in the job-restriction group. An aerobic threshold of eight METs, an abnormal EKG, an intermediate CV risk, and an abnormal CXR were the causes of restriction. The job-restriction group had a higher 10-year CV risk and higher systolic blood pressure, although these differences were not statistically significant. The wildland firefighters were unfit for their task requirements and were more at risk of cardiovascular health compared to the estimated risk of the general Thai population. To improve the health and safety of wildland firefighters, pre-placement exams and health surveillance are urgently needed.

## 1. Introduction

Firefighters are known to be high-risk workers [1]. They are subjected to high levels of physical stress at work while wearing bulky safety gear and performing demanding physical duties, such as lifting, pulling, and climbing. They confront extreme temperatures, toxic gases and substances, shift work, and psychological stress in emergencies, which are potential cardiovascular hazards [2]. Both urban and wildland firefighters encounter dangerous hazards, but their locations, circumstances, and strategies are different. Wildland firefighters could experience more unsafe conditions, such as steep terrain, rocky and muddy ground surfaces, isolated and remote areas, limited escape options, and long working hours lasting up to 14 to 21 days of assignments. They also require specialized equipment, such as piles, pulaskis, and hoes, as well as expertise skills, such as smokejumping [3]. Even though there is a standard for personal protective equipment (PPE) for wildland firefighters, it seems unlikely that they would use self-contained breathing apparatus (SCBA) in the same situation as urban firefighters [3,4]. Because SCBA is too bulky and heavy and has limited-time use, wildland firefighters are more likely to be exposed to hazardous hazards through their respiratory systems [5].

Performing firefighting tasks leads to physiological change through an increase in heart rate and prolonged beating in higher-intensity zones, which is positively correlated with the duration of fire suppression [6]. For the health and safety of firefighters, an adequate cardiopulmonary condition is necessary. Aerobic capacity is one of the best measures to determine cardiopulmonary fitness. Many countries develop their aerobic capacity standard for firefighters’ fitness for duty. For newly hired firefighters in the US, the National Fire Protection Agency (NFPA) mandates at least 12 metabolic equivalents (METs) for new recruitment and at least 8 METs for yearly examinations. Firefighters who have less than eight METS have to limit their duties and improve their physical fitness [7]. In the UK, VO2 max above 42.3 mg/kg/min is the minimum standard, and a value below 35.6 mg/kg/min is considered unfit. A firefighter who has 35.6–42.2 mg/kg/min needs fitness training [8]. In Australia’s New South Wales, candidates must have at least 12 METs; if not, they must increase their fitness [9]. Recent studies demonstrate a positive association between high levels of physical fitness and firefighters’ performance and ability [10,11]. Additionally, a low level of physical fitness raises the risk of cardiovascular disease [12]. There are also many health impacts on wildland firefighters’ cardiopulmonary status from occupational exposure [13,14,15,16,17], such as increased prevalence of hypertension and cardiovascular symptoms [14,15], and decreased lung function [16,17]. Continuous occupational exposure could produce a long-term effect on cardiopulmonary status and increase cancer risk [13]. Health impact might lead firefighters to perform their tasks in an unsafe and ineffective manner and disqualify their work. Therefore, it is crucial to offer a periodic medical examination to assess employee health and safety and ensure that firefighters can perform their full range of duties.

In Thailand, urban firefighters and wildland firefighters have somewhat different tasks, and wildland firefighters might experience more dangerous hazards due to the nature of their job. As a developing country or an upper-middle-income country, there is still no standard for pre-placement, periodic examination, and health surveillance of wildland firefighters. Without the emphasis on risk-based evaluation, only general health examinations are performed. Aerobic capacity tests, such as exercise stress tests, are rarely performed in Thailand to determine work fitness due to a lack of resources, high cost, and a lack of cardiologists to administer them. This study aims to determine the cardiopulmonary fitness level of wildland firefighters by following international standards and using practical methods. The findings may support the creation of occupational health management guidelines for Thai wildland firefighters.

## 2. Materials and Methods

### 2.1. Study Design and Population

This cross-sectional descriptive study aimed to enroll all 610 active wildland firefighters in Chiang Mai to assess their fitness levels for work. The firefighters were informed about the study through a coordination with the Organization of Protected Areas, Regional Office 16, Department of National Parks, Wildlife, and Plant Conservation, Thailand. The participants were required to have worked for at least a year and be at least 18 years old to participate in the study. This study’s inclusion criteria and items of cardiopulmonary fitness assessment were communicated to the coordinator of the organization, who then communicated the information to all wildland firefighters via organizational communication. This study was conducted from 15 December 2021 to 25 January 2022. Figure 1 represented the enrollment and discontinuation of study participants.

### 2.2. Data Collection

The fitness level for firefighting was assessed using the cardiopulmonary fitness assessment according to the NFPA standard. The wildland firefighters were interviewed by research assistants with the use of a questionnaire for their occupational history and prior history of health problems, and they were examined with calibrated tools for an assessment of their current cardiopulmonary status.

### 2.3. Questionnaire Design

The participants were interviewed using a questionnaire, which was adapted from the NFPA standard for collecting firefighting tasks and from the medical clearance form of the respiratory protection program for history of cardiopulmonary problems. This questionnaire was made up of three main parts:(1)General information of the participants, including age, gender, body weight (kg), height (cm), waist circumference (cm), body mass index (BMI, kg/m^2^), smoking status, and alcohol drinking status. BMI was categorized into four groups using the Asian BMI classification: underweight (<18.5 kg/m^2^), normal weight (18.5–22.9 kg/m^2^), overweight (23–24.9 kg/m^2^), and obese (≥25 kg/m^2^) [18].(2)Information on work tasks, including work experience (years), working hours (h/day), shift work, and job tasks.(3)Prior cardiopulmonary problems, such as myocardial infarction, arrythmia, asthma, and stroke.

### 2.4. Cardiopulmonary Fitness Assessment

This section was made up of three parts:(1)Spirometry, chest X-ray (CXR), and electrocardiogram (EKG) were performed for the participants. The spirometry test was assessed using the SpiroMaster PC-10. The procedures were performed and required at least three acceptable graphs, following the ATS/ERS standards [19]. Various parameters, including FEV1, FCV, and FEV1/FCV, were collected. The Thai Siriraj equation [20] was used as the predicted value reference.(2)Metabolic equivalents are defined as caloric consumption during an activity. One MET means caloric consumption at rest. They are used as an estimate of functional capacity, with greater METs indicating that more energy is consumed during an activity. To estimate METs for physical activities performed at work, a face-to-face interview using the global physical activity questionnaire (GPAQ) was utilized [21]. The participants were asked about their “intensity and duration” of physical activity at work and in transportation. Using the GPAQ data, the following MET values were used to determine a person’s overall energy expenditure: four METs for moderate activity and eight METs for vigorous activity. The data were analyzed as MET minutes per week based on the intensity of physical activity and duration of activity in minutes per week. We categorized the MET groups as eight METs (19,200 MET minutes per week) and twelve METs (28,800 MET minutes per week) according to the requirements in the guidance of NFPA. The calculation was based on the assumption of 8 working hours in 5 days a week.(3)The Thai CV risk score was used to estimate the 10-year incidence prediction of cardiovascular disease [22]. The parameters included age, sex, height, blood pressure, smoking status, diabetes history, and waist circumference. The researcher measured each participant’s waist circumference. The risks of over 10% were categorized as intermediate risks.

### 2.5. Definition of Fitness Level 

According to the NFPA 1582(3)’s annual examination standards guideline [7], fitness levels were divided into two groups: “Fit” meant workers who could handle all firefighting jobs without limitation, and “Job restriction” meant those who must be restricted from some firefighting activities due to their limited cardiopulmonary health. We considered a work restriction if any participants met at least one of the following criteria: under 8 METs for aerobic capacity; intermediate CV risk with SBP > 140 mmHg or DBP > 90 mmHg; FEV1 and FVC lower than 70% as predicted by the spirometry test; and any abnormal EKG or chest X-ray in category A or B (Appendix A).

### 2.6. Statistical Analysis

The difference between the fit group and the job-restriction group was analyzed using Fisher’s exact test, unpair *t*-test, and Wilcoxon rank-sum test. STATA version 16 was used for all statistical analyses. (StataCorp. 2019. Stata Statistical Software: Version 16. College Station, TX, USA: StataCorp LLC.) Statistical significance was set up at *p*-value < 0.05.

## 3. Results

With a response rate of 10.16% of all active wildland firefighters, 62 wildfire firefighters participated in the study, but only 56 performed the spirometry test. Most participants were men, with an average age of 41.66 years. The baseline characteristics are shown in Table 1. All participants had 8 h of working hours. In addition, 72.58% of participants had 12 h of shift work.

### Cardiopulmonary Fitness

The characteristics between the 54 job-restriction workers and the 8 fit-for-duty workers were not found to be statistically different. Eight firefighters were fully fit for duty by passing all cardiopulmonary parameters’ standards, and 55 firefighters needed restrictions from their firefighting tasks. Six individuals were found to have hypertension on the test day, resulting in only 56 people being examined with spirometry. All of the participants with a completed spirometry met the pulmonary function requirements. The most unmet fitness requirements were, in order, METs below eight, abnormal EKG, intermediate CV risk, and abnormal CXR. The proportion between fit and job restriction in each of the cardiopulmonary parameters is shown in Figure 2. The main cause of abnormal EKG was arrhythmia (PVC and long QTc) and ST abnormality (ischemia and pericarditis). One restriction from CXR showed subsegmental atelectasis. The overall cardiopulmonary parameters are presented in Table 2.

Of the total 54 participants with job restriction, 43 people were limited by METs alone, 7 people were limited by METs and abnormal EKG, 3 people were limited by METs and high CV risk, and 1 person was limited by METs and abnormal CXR. The subset of cardiopulmonary causes of unfitness, represented in an Euler diagram, is shown in Figure 3.

The “job restriction” group had significantly fewer MET minutes per week compared to the fit-for-duty group (*p* < 0.001). This group also had a greater ten-year CV risk and higher systolic blood pressure; however, these differences were not significant. This trend of difference between the two groups is shown in Figure 4.

## 4. Discussion

This study showed that only 12.9% of participants were entirely fit for their jobs. In addition, all 54 participants failed in cardiopulmonary fitness based on the NFPA annual aerobic capacity requirement [7]. To work without restrictions, they must maintain an aerobic capacity of at least eight METs. In comparison, the maximal oxygen consumption (VO2 max) standard proposed by the NFPA was passed by 51.0% of Colorado firefighters [23] and 27.5% of New Mexico wildland firefighters [24], which showed a higher proportion than this study. However, both studies were conducted with firefighters at a younger age. Age is a potential factor for the decline in aerobic capacity, especially at 40–50 years of age [25]. According to research conducted in Belgium [26], the percentage of firefighters who meet the 42 mL/kg/min VO2 max criterion declines at 45 years old and significantly decreases at the age of 50, with three-quarters failing this criterion at this age. Lower aerobic capacity increases cardiovascular disease [12,27] and increases the risk of injury [28]. One of the job restriction conditions was due to abnormal EKG, which showed arrhythmia and ST abnormalities. These findings might have been caused by work conditions due to firefighters being confronted with many potential cardiovascular hazards in their work, such as CO, smoke, physical work, heat stress, shift work, poor diet, and occupational stress [2]. There is evidence that fire suppression is associated with thrombus formation and vascular dysfunction, which are the pathological mechanisms for an acute myocardial infarction [29]. In one study, EKG tracing at 12 h post-fire suppression showed a rate of 20% for ventricular arrhythmia and 16% for ST-segment changes [30]. According to a cohort study of Danish firefighters, there was considerable increase in CVDs, such as angina pectoris, myocardial infarction, and atrial fibrillation, in firefighters compared to the general population when using standardized incidence ratio of 1.16, 1.16, and 1.25, respectively [31]. Even though we discovered a significant number of cardiopulmonary causes of work restrictions, with 6.45% for arrhythmia and 4.84% for ST abnormalities in our study, we could not explain if these problems were already there before being exposed to cardiovascular risk at work or if they were caused by work. In the case of high-risk job tasks, these firefighters must be restricted in their work fitness if they had a potentially life-threatening abnormal EKG or a high CV risk. There is a significant correlation between physical ability performance and cardiovascular diseases [11,12]. Screening of firefighters’ health using pre-placement examination is necessary to assess their fitness to work. 

For pulmonary fitness, four participants had a mild obstruction and one participant had a mild restrictive lung condition. The NFPA’s guidelines, however, let those with mild abnormal pulmonary function without symptoms work. Wearing SCBA reduces risk exposure but increases physiological strain by increasing airflow resistance, dead space, and breathing rate, which could limit a firefighter’s ability in case of pulmonary disease [32,33]. In case of health effects, hazards at work might also harm or aggravate the respiratory system in firefighters [5]. Systematic review studies report a significant decline in pulmonary function both in cross-shift and cross-season firefighters [13]. A longitudinal study of spirometry found that FEV1, FVC, and PEF decline after smoke exposure and significant decline at more than 24 h post-exposure. FCV and FEV1 have been shown to decrease daily when compared to the prescribed burn period [17]. A previous study conducted in Chiang Mai reported that another group of firefighters had short-term effects on small airway dysfunction but no effect on spirometry [34]. This might differ from the participants in this study, as their average age was lower than that of the participants in this study. Wildland firefighters mostly do not wear SCBA due to inappropriate movement and flexibility, which can expose them to more hazards to their lungs. Thus, even though all participants met pulmonary function standards, health surveillance to follow lung decline in firefighters should be performed. To avoid work-related cardiopulmonary disease in Thai wildland firefighters, pre-placement examinations, medical monitoring, and health screening are required.

Some wildland firefighters’ organizations in Chiang Mai conduct physical fitness testing by having candidates run five kilometers and perform one minute of pushups to determine their physical fitness. Although their organizations conduct these periodic physical fitness evaluations to evaluate their fitness and estimate their readiness for job tasks, the proportion of physiologically fit individuals remains low. Low aerobic capacity could be improved through physical training programs and an increase in physical activity [11,35,36,37]. A meta-analysis study showed that an exercise intervention, including aerobic exercise, resistance exercise, or a combined method of 3–4 sessions/weeks for 16.5 ± 10 weeks, could improve aerobic capacity [35]. Chizewski et al. [11,37] reported that high-intensity fitness training improved fitness and firefighting ability [37]. After performing a fitness training program for seven weeks, VO2 max increased from 40.84 ± 5.09 in the first week to 45.30 ± 5.24 in the seventh week [11]. Therefore, even though cardiopulmonary fitness declines with age, it might be improved or maintained in a good condition by a fitness training program. The training program of one study encourages firefighters to engage in moderate-to-vigorous leisure-time physical activity and to improve cardiovascular workload while performing job tasks [36].

In developed countries, including the United States of America (USA), the United Kingdom (UK), and Australia, there are standards for firefighters’ fitness for duty. All workers are required to have a pre-placement examination for fitness evaluation and baseline health surveillance, as well as an annual examination to determine if there are any work-related hazards. To the best of our knowledge, Thailand still needs to reach a consensus on a fit-for-duty standard for firefighters. The majority of the findings on work restrictions provide good supporting evidence to emphasize the urgency of developing occupational health management standards for Thailand in the near future. Urban and wildland firefighters’ pre-employment and routine checkup programs are determined by organizational management, which could vary depending on the setting. Available resources and national contexts might be considered to develop guidelines for fit-to-work and health surveillance among Thai firefighters. For example, the exercise stress test, as a gold standard for aerobic capacity, is a limited resource for many developing countries. Non-exercise tools have been developed to estimate VO2 max, which have more practical use in large populations. Self-perception of aerobic fitness, such as the physical activity questionnaire, has been widely used to determine firefighters’ aerobic capacity [38,39]. A study with 102 working-age adults reported that self-reported daily vigorous physical activity was associated with aerobic capacity [38]. As a beginning for developing a standard in a country with limited resources, a self-perception questionnaire might be useful for screening. Utilizing screening or non-invasive methods might aid in the early detection of any abnormalities in the cardiovascular system.

To our best knowledge, this is the first investigation of cardiopulmonary fitness in Thai wildland firefighters. There are several limitations to this study. First, as a cross-sectional study, the direction of associations cannot be determined. This study only evaluated cross-sectional health status without comparing it to the baseline because there were no baseline data for each participant and also because of an inadequate pre-placement examination program. Second, there might have been selection bias due to the low number of wildland firefighters who agreed to participate in this study. Although about 10% of the study population responded, the findings could be considered generalizable to other firefighting settings with different contexts and resources. However, the majority of fire-prone regions in Thailand have a similar geography. Moreover, most of the wildland firefighters in Thailand are Thai nationals with comparable job descriptions. The generalizability of this study’s findings to other Thai wildland firefighters could be advantageous. Another factor could contribute to selection bias. As we attempted to recruit all Chiang Mai wildland firefighters, we recognized that people who might be interested in participating were those who were concerned about their health. This might have led to a higher average age of the participants compared to other studies. Third, as the MET evaluation used in this study was based on self-report using the GPAQ for practical assessment, it might not reflect the firefighters’ actual aerobic capability as accurately as quantitative methods. Future studies should be conducted, including a longitudinal study with baseline health assessments to determine the causal relationships and trends in participants’ fitness. We also suggest using a more effective recruitment method to increase the number of participants in future studies.

## 5. Conclusions

The majority of Thai wildland firefighters had low cardiopulmonary fitness according to international standards. This might affect their work productivity and health. Low aerobic capacity was the major cause of job restriction, and the job-restriction group tended to have a higher risk of CVD risk, even though this trend was not statistically significant. These results demonstrate that Thailand needs an effective surveillance system for hazardous occupations. To protect wildland firefighters from potential dangers, it is urgent to establish standards, guidelines, and policies for fitness for duty. Every wildland firefighter should receive training to improve their functional fitness, and non-invasive screening techniques could be a useful tool for monitoring their fitness.

## Figures and Tables

**Figure 1 ijerph-20-03527-f001:**
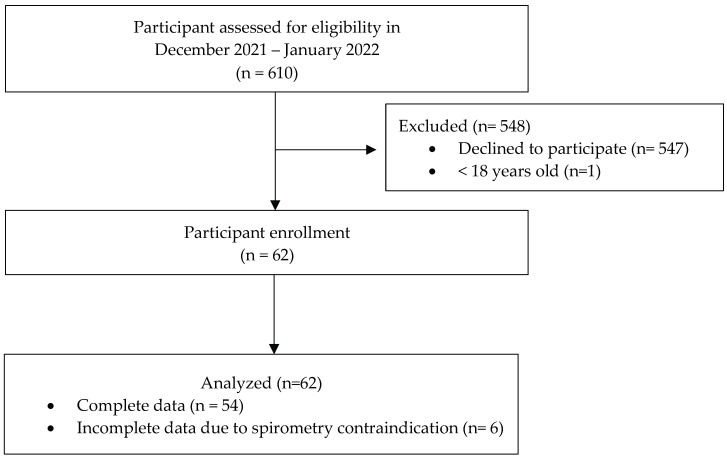
The Study flow diagram.

**Figure 2 ijerph-20-03527-f002:**
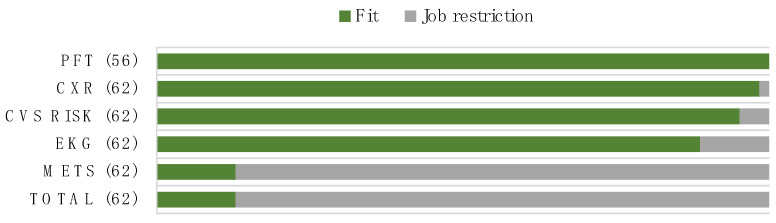
Fit-for-duty status among wildland firefighters based on cardiopulmonary parameters.

**Figure 3 ijerph-20-03527-f003:**
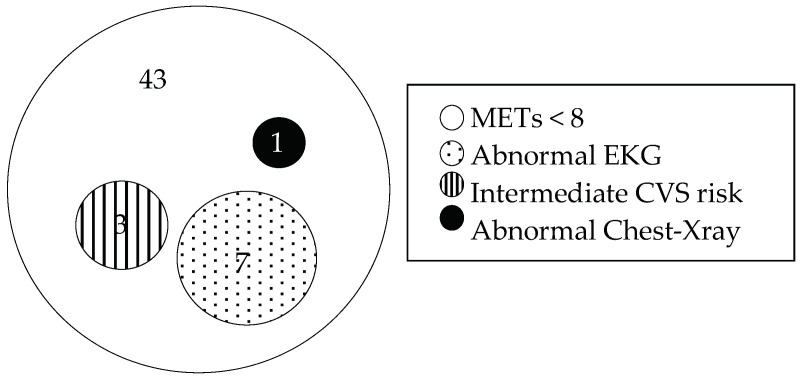
Causes of job restriction in wildland firefighters (*n* = 54).

**Figure 4 ijerph-20-03527-f004:**
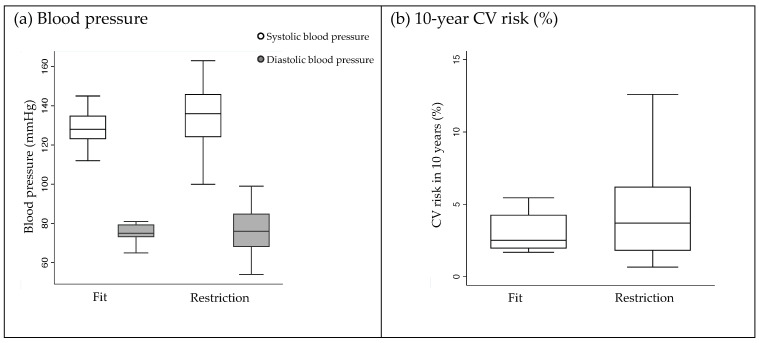
Risk factors of cardiovascular diseases between the fit-for-duty and job-restriction groups. (**a**) The box plot shows that the job-restriction group has higher systolic blood pressure than the fit--for-duty group, even though the difference is not statistically significant. (**b**) The box plot shows the risk of developing a CVD within ten years is higher in the job-restriction groups than in the fit-for-duty group. However, the difference is not significant.

**Table 1 ijerph-20-03527-t001:** Baseline characteristics of wildfire firefighters and characteristics between the fit-for-duty and job-restriction groups.

Characteristics	Total (*N* = 62)	Fit (*n* = 8)	Job Restriction (*n* = 54)	*p*-Value
	*N* (%)	*n* (%)	*n* (%)	
Sex				0.656 *^a^*
Male	59 (95.16)	8 (100)	51 (94.44)	
Female	3 (4.84)	0	3 (5.56)	
Age (year), mean (SD)	41.66 (10.42)	42.63 (6.76)	41.52 (10.90)	0.700 *^b^*
Weight (kg), mean (SD)	67.39 (9.86)	67.63 (5.45)	67.35 (10.39)	0.911 *^b^*
Height (cm), mean (SD)	166.24 (6.91)	166.88 (5.22)	166.15 (7.16)	0.731 *^b^*
Waist circumference (cm), mean (SD)	34.75 (3.42)	36.10 (2.32)	34.55 (3.53)	0.129 *^b^*
Work experience (year), mean (SD)	13.23 (9.97)	16 (8.09)	12.82 (10.22)	0.348 *^b^*
BMI (kg/m^2^), mean (SD)	24.30 (3.06)	24.30 (1.73)	24.30 (3.22)	0.996 *^b^*
BMI categories				1.000 *^a^*
Underweight	1 (1.61)	0	1 (1.85)	
Normal	18 (29.03)	2 (25.00)	16 (29.63)	
Overweight	22 (35.48)	3 (37.50)	19 (35.19)	
Obese	21 (33.87)	3 (37.50)	18 (33.33)	
Shift work				0.620 *^a^*
Yes	45 (72.58)	6 (75.00)	39 (72.22)	
No	17 (27.42)	2 (25.00)	15 (27.78)	
Smoke				0.712 *^a^*
Active smoker	26 (41.94)	4 (50.00)	22 (40.74)	
Ex-smoker	20 (32.26)	3 (37.50)	17 (31.48)	
Non-smoker	16 (25.81)	1 (12.50)	15 (27.78)	
Alcohol				0.403 *^a^*
Regular drinking	32 (51.61)	3 (37.50)	29 (53.70)	
Social drinking	16 (25.81)	4 (50.00)	12 (22.22)	
Ex-drinking	5 (8.06)	0	5 (9.26)	
Non-drinking	9 (14.52)	1 (12.50)	8 (14.81)	

*^a^* = Fisher’s exact test, *^b^* = unpair *t*-test, and BMI = body mass index.

**Table 2 ijerph-20-03527-t002:** Comparing the fit-for-duty and job-restriction groups in terms of cardiopulmonary fitness parameters.

Cardiopulmonary Parameters	Total (*N* = 62)	Fit (*n* = 8)	Job Restriction (*n* = 54)	*p*-Value
	mean (SD)	mean (SD)	mean (SD)	
METs (minutes per week)	7333.55 (8755.27)	24,240 (5940) (median, (IQR))	4640 (5241.20)	<0.001 *^a^*
METs				<0.001 *^b^*
MET > 12	2 (3.22)	2 (25.00)	0	
MET 8–12	6 (9.68)	6 (75.00)	0	
MET < 8	54 (87.10)	0	54 (100)	
CV risk in 10 years (%)	4.53 (3.54)	2.51 (2.33) (median, (IQR))	4.68 (3.66)	0.629 *^a^*
CV risk classification (*n*, %)				1.00 *^b^*
Low risk	57 (91.93)	8 (100)	49 (90.74)	
Intermediate risk	5 (8.07)	0	5 (9.26)	
Resting heart rate (bpm)	88.48 (13.51)	88 (11.03)	88.56 (13.92)	0.901 *^c^*
BMI (kg/m^2^)	24.30 (3.06)	24.30 (1.73)	24.30 (3.22)	0.996 *^c^*
SBP (mmHg)	134.35 (15.87)	128.63 (10.35)	135.20 (16.43)	0.231 *^c^*
DBP (mmHg)	76.58 (10.80)	76.75 (8.36)	76.56 (11.18)	0.875 *^c^*
Pulmonary function *				
FEV_1_ (mL)	3425.54 (811.37)	3316.25 (495.58)	3443.75 (855.24)	0.805 *^c^*
FVC (mL)	4171.79 (939.73)	4116.25 (604.69)	4181.04 (989.11)	0.561 *^c^*
Pulmonary function result * (n, %)				1.000 *^b^*
Normal	51 (91.07)	8 (100)	43 (89.58)	
Obstruction	4 (7.14)	0	4 (8.34)	
Restriction	1 (1.79)	0	1 (2.08)	
Underlying disease (n, %)				
None	53 (85.48)	7 (87.50)	46 (85.19)	
Myocardial infarction	1 (1.61)	0	1 (1.85)	1.000 *^b^*
Arrhythmia	2 (3.23)	0	2 (3.70)	1.000 *^b^*
Asthma	4 (6.45)	1 (12.50)	3 (5.56)	0.433 *^b^*
Stroke	2 (3.23)	0	2 (3.70)	1.000 *^b^*

* *n* = 56, *^a^* = Wilcoxon rank-sum test, *^b^* = Fisher’s exact test, *^c^* = unpair *t*-test, METs = metabolic equivalent, CV = cardiovascular, BMI = body mass index, SBP = systolic blood pressure, DBP = diastolic blood pressure, FEV1 = forced expiratory volume in 1 s, and FVC = forced vital capacity.

## Data Availability

The data presented in this study are available from the corresponding author upon request.

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
