# Peer review of "The Urgent Need for Cardiopulmonary Fitness Evaluation among Wildland Firefighters in Thailand"

_ijerph, 2023, doi:10.3390/ijerph20043527_

Round 1
Reviewer 1 Report
Important topic... novel data
Despite limitations which include small sample, low response, rate, laack of formal fitness testing; the data are important to share and the authors highlight differences in their sample with those published in other countries
The abstract is really important and does not give a good impression of this article.. Therefore my 1st comments below illustrate some substantive problems that the reader would have when approaching this abstract
Please check the tense used throughout your manuscript. For example, the 1st line of your abstract says that Wildland firefighters were one of the highest-risk occupation----are is correct
“ non-laboratory cardiovascular risk assessment” is not a useful way to define your assessment because you are defining it by what it is not rather than what it is… Be clear about is it submaximal or maximal test.. What is the assessment criteria everyone knows what a Vo2 max test is but no one will be clear about what a nonlaboratory test is
Eighty-seven percent of participants were in the work-restriction group. An aerobic threshold of 8 METs, an abnormal EKG, intermediate CV risk, and an abnormal CXR were the causes of restriction…. This is an extraordinary level of work restriction where the majority of the working population actually is defined as having a work restriction.. Either there is something wrong with the screening to become a wildland firefighter or the training or the definition of work restriction.
Fisher Exact and Wilcoxon rank-sum tests were used to compare cardiopulmonary parameters….. With large sample sizes I would’ve expected a normal distribution and parametric statistics
The wildland firefighters were less fit for their task requirements and more at risk of cardiovascular health…. Less fit than who?…. There is not a clear match between your study design which was cross-sectional and some of your statements for example how can you be stating the 10 year risk when you have a cross-sectional design…
There is a reliance on P values rather than stating effect sizes. P values really mean nothing especially with large sample sizes and the American statistical Association has discouraged the use of P values for making conclusion. The size and the size and direction of the effect is what matters for Association and the effect size is what matters for differences.
Main paper
More info needed about the data collection on To estimate METs for physical activities performed at work, a face-to-face inter-116 view using the global physical activity questionnaire (GPAQ) was utilized [14]. The data 117 were analyzed to MET minute per week by the intensity of physical activity and duration 118 of activity in minutes per week. We determined the METs group by 8 METs (19,200 MET 119 minutes per week) and 12 METs (28,800 MET minutes per week) according to the require-120 ments in the guidance of NFPA. The calculation was assumed by 8 working hours in 5 121 days a week.
Do ff work 8 hour day ? is this valid given context of they work?
When classifying fit or unfit the cut-off determines how people fall on one side or the other…Cut off for each parameter were un-131 der 8 METs for aerobic capacity, intermediate CV risk with SBP > 140 mmHg or DBP > 90 132 mmHg, FEV1 and FVC lower than 70% predicted for spirometry test, and any abnormal 133 EKG and Chest X ray in category A or B (appendix A). Say more about how this was selected and if valid – important given the high percentage considered unfit
Analysis - again it seems like nonparametric not the best choice for this data.no rationale provided; this should be normally distributed
Response of 10% not surprising given tests but potentially only those with concerns came in so maybe biased
Age is a confounder here and should be clearly managed and discussed… how age related was risk? How many of those classified at risk were mainly age related versus findings on exam … ..why do you think your sample was older than other studies- is this policy related?
Author Response
To the reviewer 1,
We would like to thank you for your useful comments. We revised our manuscript in response to your suggestions and listed each change individually. We believe that the revised manuscript is now greatly improved after the revisions have been made.
even though our native English consultant continues to proof the whole manuscript, which may not finish in time. We revised other points and submitted an updated manuscript to the journal without re-proofreading the English. If we get the English-proofed manuscript, we will send the completed version to the journal immediately.
Best regards,
Pheerasak Assavanopakun

Reviewer 2 Report
I have carefully reviewed manuscript of Panumasvivat et al. titled: " The Urgent Need for Cardiopulmonary Fitness Evaluation among Wildland Firefighters in Thailand ".
Please check the comments for your manuscript:
• Introduction
o Line 30 - Firefighter is known to be high-risk workers – citation for this? Prove why is this known as high-risk work.
o Please try to find flow in introduction with problem and with your hypothesis or your vision of a problem which you want to solve – try to get direct into problem and rationale after good background base
o Do you have hypothesis?
o Update your introduction with new references, as only in introduction you have 2 newly published references.
• Methods
o Why have so many participants refused to enrol into the study?
o Why did not you try to solve this with their leadership with contract or similar? Because it is pity you have so big sample but a lot of them refused
o You should have in text also some basic descriptive parameters of firefighters (participants)
• Results
o Figure 2 – explain better fit / job restriction – readers will not know what is this about
• Discussion
o What are the strengths of this study?
o What are practical implications?
o What do authors suggest for new studies?
o Other parts of discussion are written well according to the results you have obtained from the study
• Conclusions
o Conclusion is weak…write about the exact results of the study and reorganise this with exact conclusions and not already written phrases.
• References
o You can’t have only 7 references out of 31 in the last 3 years
o Correct references – several are not correctly written. Especially the ones that refer to other sources than journals

Author Response
To the reviewer 2,
We would like to thank you for your useful comments. We revised our manuscript in response to your suggestions and listed each change individually. We believe that the revised manuscript is now greatly improved after the revisions have been made.
even though our native English consultant continues to proof the whole manuscript, which may not finish in time. We revised other points and submitted an updated manuscript to the journal without re-proofreading the English. If we get the English-proofed manuscript, we will send the completed version to the journal immediately.
Best regards,
Pheerasak Assavanopakun

Reviewer 3 Report
The urgent need for cardiopulmonary fitness evaluation among wildland firefighters in Thailand
The overall health and fitness for duty of fire fighters is an important public health issue.
Overall, there are a significant number of grammatical errors, and the language is sometimes unclear, requiring multiple reads or usage of context clues. This diminishes the ability to evaluate the study from a scientific point of view. For example, the first sentence of the abstract reads: Wildland firefighters were one of the highest risk occupations.” Using the verb implies that they are no longer high risk. Firefighters are people, not occupations. What is the risk that is high? Is it injury? Is it cancer? Is it divorce? Other?
Low Participation: The authors should discuss more about the low participation (62 of 610). They should provide any information known about the entire group. More should be provided in the limitation section of the discussion. Do they have any insight as to why the firefighters did not participate? This would be important to address if more studies are to be done on this population. Since so many of the participants were “no fit” is it possible that they wanted to be examined? Conversely, was there any downside to participate and be labeled “job restriction?
Introduction: The use of METs as a mandate is unclear. This should be better explained. See methods.
Line 74. What are practical methods? Xrays, spirometry and EKG seem fairly technical.
Methods.
Line 84. Don’t include results in the methods.
Line 103. Was waist circumference collected via questionnaire? Is this something the firefighers would know?
METS. How did the authors get METS from GPAQ? What is the unit? The authors discuss minutes per week (but should define earlier as a 40 hour work week). How are non-work activities included?
CV risk score. How was this determined? What does it mean to be “over 10%”?
Results.
Fit vs. job restriction is a blurry definition. The list in the methods isn’t clear if any of the factors or all factors are restrictive.
Table 1. % for BMI are missing. The BMI categories should be defined. Ranges are appreciated for continuous variables.
Discussion. In general, it is not always clear if the authors are discussing the study population, Thailand or other countries.
Author Response
To the reviewer 3,
We would like to thank you for your useful comments. We revised our manuscript in response to your suggestions and listed each change individually. We believe that the revised manuscript is now greatly improved after the revisions have been made.
even though our native English consultant continues to proof the whole manuscript, which may not finish in time. We revised other points and submitted an updated manuscript to the journal without re-proofreading the English. If we get the English-proofed manuscript, we will send the completed version to the journal immediately.
Best regards,
Pheerasak Assavanopakun

Round 2
Reviewer 2 Report
Thank you for corrections. Although I think you need to add more references.
Author Response
Responses to the Reviewer 2 Comments (round 2)
Point 1: Thank you for corrections. Although I think you need to add more references.
Response 1: We would like to thank you, reviewer 2, for the suggestion. We added more references to our manuscript as per your suggestion. We believe that the revised manuscript is now greatly improved after the revisions have been made.
- We added the references in the introduction to increase support for wildland firefighters' duties and health risks. It nows reads
“Even though there was a standard for personal protective equipment (PPE) for wildland firefighters, it seemed unlikely that they would use self-contained breathing apparatus (SCBA) in the same situation as urban firefighters [1,2]. Because SCBA was too bulky, heavy, and limited-time use, so wildland firefighters were more likely to be exposed to hazardous hazards through their respiratory systems [3].” (Line 40-45, Page 1)
“In case of health effects, Hazards at work might also harm or aggravate the respiratory system in firefighters [3].” (Line 243-244, Page 8)
- We added the reference to the discussion in order to compare pulmonary effects with another population studied in a subject area. It nows reads
“A previous study conducted in Chiang Mai reported that another group of firefighters had short-term effects on small airway dysfunction but no effect on spirometry [4]. This may differ from the participants in this study, as their average age was lower than that of the participants in this study.” (Line 248-252, Page 8)
References
- The Department of the Interior. The Essential Functions and Work Conditions of a Wildland Firefighter. Available online: https://www.nifc.gov/medical_standards/index.html. n.d. (accessed on 5 January 2023).
- The Department of the Interior. Federal Interagency Wildland Firefighter Medical Standards. Available online: https://www.nifc.gov/medical_standards. n.d. (accessed on 5 January 2023).
- Adetona, O.; Reinhardt, T.E.; Domitrovich, J.; Broyles, G.; Adetona, A.M.; Kleinman, M.T.; Ottmar, R.D.; Naeher, L.P. Review of the health effects of wildland fire smoke on wildland firefighters and the public. Inhal Toxicol 2016, 28, 95-139, doi:10.3109/08958378.2016.1145771.
- Niyatiwatchanchai, N.; Pothirat, C.; Chaiwong, W.; Liwsrisakun, C.; Phetsuk, N.; Duangjit, P.; Choomuang, W. Short-term effects of air pollutant exposure on small airway dysfunction, spirometry, health-related quality of life, and inflammatory biomarkers in wildland firefighters: a pilot study. Int J Environ Health Res 2022, 10.1080/09603123.2022.2063263, 1-14, doi:10.1080/09603123.2022.2063263.